# Structural insights into the light-driven auto-assembly process of the water-oxidizing $Mn_4CaO_5$-cluster in photosystem II

Miao Zhang[1†], Martin Bommer[1†‡], Ruchira Chatterjee[2], Rana Hussein[1], Junko Yano[2], Holger Dau[3*], Jan Kern[2*], Holger Dobbek[1*], Athina Zouni[1*]

[1]Institut für Biologie, Humboldt-Universität zu Berlin, Berlin, Germany; [2]Molecular Biophysics and Integrated Bioimaging Division, Lawrence Berkeley National Laboratory, Berkeley, United States; [3]Freie Universität Berlin, Berlin, Germany

**Abstract** In plants, algae and cyanobacteria, Photosystem II (PSII) catalyzes the light-driven splitting of water at a protein-bound $Mn_4CaO_5$-cluster, the water-oxidizing complex (WOC). In the photosynthetic organisms, the light-driven formation of the WOC from dissolved metal ions is a key process because it is essential in both initial activation and continuous repair of PSII. Structural information is required for understanding of this chaperone-free metal-cluster assembly. For the first time, we obtained a structure of PSII from *Thermosynechococcus elongatus* without the $Mn_4CaO_5$-cluster. Surprisingly, cluster-removal leaves the positions of all coordinating amino acid residues and most nearby water molecules largely unaffected, resulting in a pre-organized ligand shell for kinetically competent and error-free photo-assembly of the $Mn_4CaO_5$-cluster. First experiments initiating (i) partial disassembly and (ii) partial re-assembly after complete depletion of the $Mn_4CaO_5$-cluster agree with a specific bi-manganese cluster, likely a di-μ-oxo bridged pair of Mn(III) ions, as an assembly intermediate.

*For correspondence: holger. dau@fu-berlin.de (HDa); jfkern@ lbl.gov (JK); holger.dobbek@hu-berlin.de (HDo); athina.zouni@hu-berlin.de (AZ)

†These authors contributed equally to this work

Present address: ‡Max-Delbrück-Center for Molecular Medicine, Berlin, Germany

Competing interests: The authors declare that no competing interests exist.

## Introduction

The global oxygen, carbon and nitrogen cycles are driven by complex metalloenzymes. In oxygenic photosynthesis, carried out by plants, algae and cyanobacteria, transformation of light into chemical energy takes place in Photosystem I and Photosystem II (PSII). The latter harbors the water-oxidizing complex (WOC), which is responsible for oxygen evolution (*McEvoy and Brudvig, 2006*; *Barber, 2009*; *Dau et al., 2012*; *Cox and Messinger, 2013*; *Yano and Yachandra, 2014*; *Shen, 2015*). In the last decade, X-ray crystal structures of the dimeric PSII core complexes (dPSIIcc) from the thermophilic cyanobacteria *Thermosynechococcus elongatus* (*T. elongatus*) (*Zouni et al., 2001*; *Ferreira et al., 2004*; *Loll et al., 2005*; *Guskov et al., 2009*; *Hellmich et al., 2014*) and *Thermosynechococcus vulcanus* (*T. vulcanus*) (*Umena et al., 2011*; *Suga et al., 2015*) showed that each monomer contains at least 20 protein subunits including nearly 100 cofactors and an inorganic metalcluster, designated as the $Mn_4CaO_5$-cluster, which catalyzes the oxidation of two molecules of water yielding molecular oxygen, four protons and four 'energized' electrons (*McEvoy and Brudvig, 2006*; *Barber, 2009*; *Dau et al., 2012*; *Cox and Messinger, 2013*; *Yano and Yachandra, 2014*; *Shen, 2015*). The electrons are transferred from water to the final electron acceptor, a mobile plastoquinone, called $Q_B$ (*Müh et al., 2012*). In cyanobacteria, three membrane-extrinsic subunits are located at the lumenal side of the thylakoid membrane: PsbO (33 kDa), PsbV (cyt c-550) and PsbU

(12 kDa) (*Figure 1B*). They interact with intrinsic membrane proteins and stabilize the $Mn_4CaO_5$-cluster (*Bricker et al., 2012*; *Nickelsen and Rengstl, 2013*).

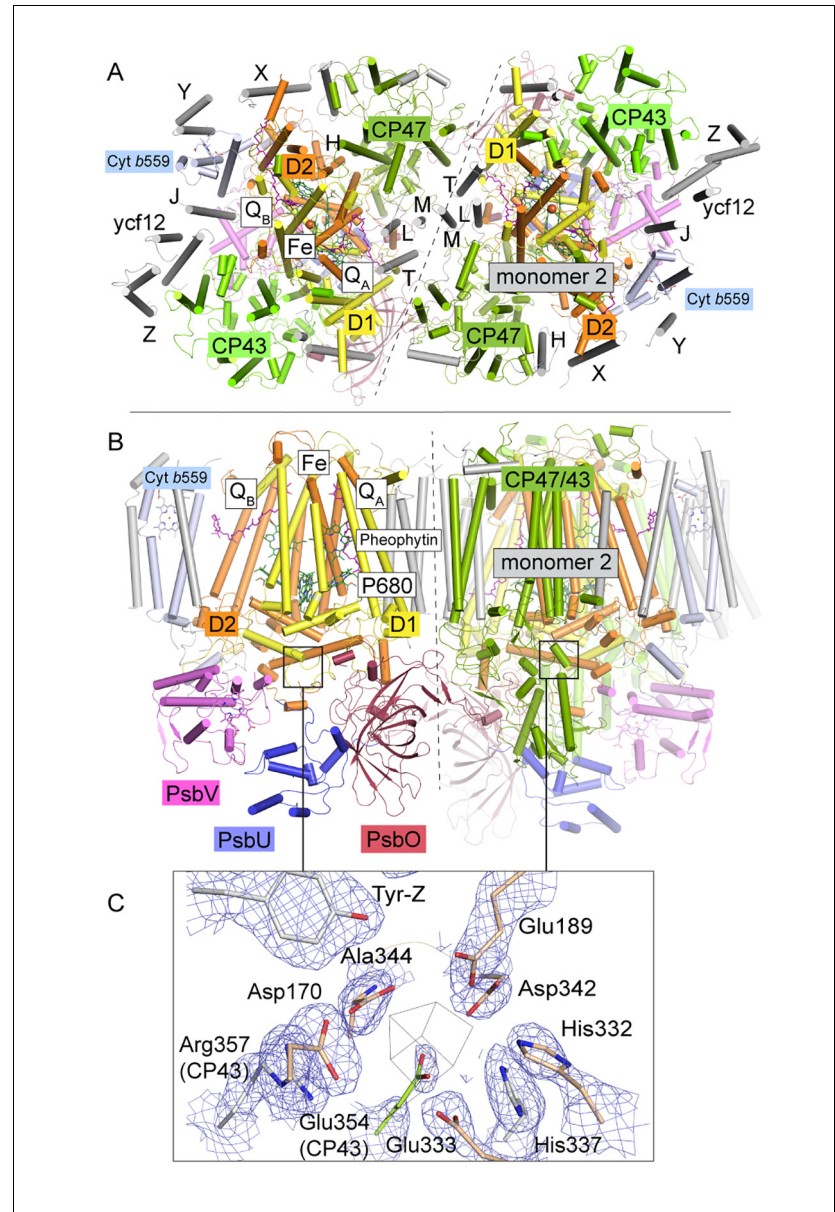

**Figure 1.** *T. elongatus* PSIIcc dimers viewed. (**A**) from the cytoplasmic side and (**B**) from the membrane. The reaction centre transmembrane subunits D1/D2, the internal antennae (CP43/CP47, omitted in the left monomer) and the membrane extrinsic subunits PsbU/V/O are highlighted in color. Small transmembrane subunits comprising a single α-helix (two for PsbZ) are indicated by single letter referring to the PsbH-Y proteins and ycf12. $Q_{A/B}$ denote the plastoquinone cofactor, Fe the non-heme iron. The site of the water-oxidizing complex (WOC, $Mn_4CaO_5$) within each monomer is shown by the black box. (**C**) Electron density map ($1\sigma$ $2F_o$-$F_c$) at the depleted WOC site (shown for the locked dimer, discussed below). Free coordinating residues from the D1 subunit (cream) and CP43 (green) are shown alongside Arg357(CP43) and His 337(D1), which are within hydrogen-bond distance to cluster oxygen atoms in *Umena et al. (2011)*, (PDB:3WU2).

The following figure supplement is available for figure 1:

**Figure supplement 1.** Effect of $NH_2OH$-EDTA treatment performed on dPSIIcc solutions.

At present, the atomic-resolution structure at 1.9 Å of dPSIIcc from *T. vulcanus* enables a detailed geometry of the $Mn_4CaO_5$-cluster (*Umena et al., 2011*; *Tanaka et al., 2017*). The shape of the $Mn_4CaO_5$-cluster was described as a 'distorted chair': A $Mn_3CaO_4$ heterocubane cluster ('chair base') with an additional Mn (Mn4) in *exo* position connected to the cubane by two µ-oxo-bridges ('backrest') (*Umena et al., 2011*). Further, four water molecules are directly ligated to the $Mn_4CaO_5$-cluster, two (W1, W2) at Mn4 and the other two (W3, W4) at $Ca^{2+}$. The whole cluster is coordinated by six amino acids from the D1 subunit (PsbA) and one from the CP43 protein (PsbC).

The mechanism of the PSII repair is important to determine plant productivity and, consequently, a hot spot of physiological research with a research intensity clearly exceeding that of basic research on photosynthetic water oxidation. In the recent research, different models about this mechanism have been proposed (*Järvi et al., 2015*). *Nickelsen and Rengstl (2013)* have compared two de novo assembly models for both plants and cyanobacteria exhibiting the common main disassembly phases during the repair (see reviews for more details [*Nixon et al., 2010*; *Becker et al., 2011*; *Bao and Burnap, 2016*; *Heinz et al., 2016*]). For the light-driven assembly of the $Mn_4CaO_5$-cluster (photo-activation), its mechanism has remained elusive, despite the recent progress in the structure elucidation and mechanistic PSII research in general. The process of photo-activation involves the stepwise incorporation of the Mn and Ca ions into the 'apo-PSIIcc', that is, the dimeric PSII core complex without any metal ions bound at the WOC site. The initial steps of the photo-activation process can be illustrated in the so-called two quantum model (*Cheniae and Martin, 1971*; *Dasgupta et al., 2008*; *Becker et al., 2011*; *Bao and Burnap, 2016*): First, a Mn(II) ion binds in the dark at the high-affinity side of the apo-PSIIcc and is oxidized after absorption of a first light quantum forming an unstable Mn(III) intermediate. This step is followed by a light-independent rearrangement. Then a second light quantum drives the oxidation of a second Mn(II) ion resulting in a next assembly intermediate, a binuclear Mn complex, possibly involving two di-µ-oxo bridged Mn(III) ions (*Barra et al., 2006*). The incorporation of the other two missing Mn ions to complete the metal-cluster has not been kinetically resolved.

In this study, we focus on the determination of the structure of apo-PSIIcc. We used *T. elongatus* PSII crystals described in *Hellmich et al. (2014)*, where octaethyleneglycolmonododecylether ($C_{12}E_8$) detergent-solubilized PSII is crystallized followed by a partial removal of the detergent. This structure consists of a CP43/CP47 and PsbO interlinked row of dimers, packing akin to that in the native thylakoid membrane. The complete depletion of the $Mn_4CaO_5$-cluster was achieved by employing a mixture of $NH_2OH$ as a reducing agent of the Mn-ions and ethylenediaminetetraacetic acid (EDTA), a chelator for $Mn^{2+}$ as well as $Ca^{2+}$ ions. We performed a systematic study of determining optimal conditions for a complete depletion of the $Mn_4CaO_5$-cluster in dPSIIcc solutions and crystals, by using electron paramagnetic resonance (EPR) spectroscopy. Interestingly, the incubation of dPSIIcc with $NH_2OH$/EDTA in solution not only caused the loss of all the three extrinsic subunits (PsbU, PsbV and PsbO), but also a dissociation of dimeric into monomeric PSIIcc. These results motivated us to develop a new protocol for depletion of the $Mn_4CaO_5$-cluster in $C_{12}E_8$-dPSIIcc crystals. For the first time, we obtained a crystal structure of PSII fully depleted of the $Mn_4CaO_5$-cluster, at 2.55 Å resolution (*Figure 1*). Surprisingly, we found that all 20 protein subunits and cofactors remained largely unaffected in the absence of the metal-cluster. Small changes in apo-PSIIcc are limited to residues in the vicinity of the cluster and a destabilization of the PsbU subunit. Based on the apo-PSIIcc, initial experiments aiming for a reconstitution of the $Mn_4CaO_5$-cluster were pursued in this study, that paves the road for future structural investigation of assembly intermediates.

## Results and discussion

### $Mn_4CaO_5$-cluster depletion performed on dPSIIcc solutions

We investigated $Mn_4CaO_5$-cluster depletion of PSII in solution using various combinations of $NH_2OH$ and/or EDTA, as suggested in earlier studies (*Cheniae and Martin, 1971*; *Sugiura and Inoue, 1999*). When we treated dPSIIcc solutions with $NH_2OH$, they lost $O_2$ evolution activity (*Supplementary file 1*). In the dark-stable $S_1$-state of PSII, the WOC is a high valent Mn complex ($Mn^{III}_2 Mn^{IV}_2$). The addition of $NH_2OH$ reduces the Mn ions to $Mn^{2+}$, which is no longer stably bound to the PSII apo-protein. Thus, the $Mn_4CaO_5$-cluster is lost, inactivating light-driven $O_2$ evolution. A combination of $NH_2OH$ and EDTA in dPSIIcc solutions caused dissociation of all three extrinsic

subunits, PsbU, PsbV and PsbO, whereas all other protein subunits remained bound to PSII (*Figure 1—figure supplement 1A* for larger PSII subunits; *Supplementary file 2* for smaller PSII subunits). In addition, the absence of PsbO could destabilize the monomer-monomer interaction in the PSII core dimer (*Boekema et al., 2000*; *Komenda et al., 2010*), leading to a monomerization (*Figure 1—figure supplement 1B*). Consequently, a $NH_2OH$/EDTA treated solution of dPSIIcc in $C_{12}E_8$ lacks the extrinsic subunits mediating crystal contacts in all known high-resolution dPSIIcc crystals.

To further understand the disassembly process, we also treated the dPSIIcc solution only with EDTA at 50 mM. Under this condition, only the PsbU subunit was removed. This result indicates that PsbU is comparatively loosely bound in the PSIIcc. A destabilization of PsbU is also confirmed by the structural data discussed in the later section.

## $Mn_4CaO_5$-cluster depletion performed on $C_{12}E_8$ dPSIIcc microcrystals

The optimal concentration of $NH_2OH$/EDTA was determined by EPR studies. The results show that >30 mM $NH_2OH$ are needed in $C_{12}E_8$ dPSIIcc microcrystals to reduce all the Mn ions within the WOC (*Figure 2—figure supplement 1C*), in contrast to PSII in solution, where 20 mM $NH_2OH$ are sufficient (*Figure 2—figure supplement 1A and B*). In the following experiments, we use 50 mM $NH_2OH$ as the final concentration (*Figure 2*, purple trace). Compared to the spectrum of 'free' $Mn^{2+}$ ($MnCl_2$ solution; *Figure 2*, green trace), the $Mn^{2+}$ signal resulting from reduction of the $Mn_4CaO_5$ complex by $NH_2OH$ has narrower bands and shows modified hyperfine features. In the symmetric hexaaquo coordination in solution, $[Mn(H_2O)_6]^{2+}$, we observe a typical six-line EPR signal that arises from the interaction of the electron spin with the $^{55}Mn$ nuclear spin (I = 5/2) such that each line represents five almost degenerate spin transitions. In cases where $Mn^{2+}$ is bound in an asymmetric environment, like in a protein, a broader signal is observed due to the separation of the different $m_s$ levels, and the orientation-dependence of the transitions among them. However, in some cases (*Figure 2*, purple trace) only one of the transitions ($m_s$ 1/2 ⟷ −1/2) remain resolvable and we observe a six-line EPR signal with reduced intensity and narrower spectral width compared to that of the $[Mn(H_2O)_6]^{2+}$ (*Figure 2*, green trace). Further, washing with 50 mM EDTA ensures that all the $Mn^{2+}$ and $Ca^{2+}$ ions inside PSII are removed as evident from the red trace in *Figure 2* that shows a $Mn^{2+}$ free EPR spectrum. (It should be noted that the characteristic $TyrD^{\bullet}$ signal of EDTA washed dPSIIcc solutions is significantly weaker than the signal of the untreated sample since the $TyrD^{\bullet}$ radical is reduced after addition of $NH_2OH$.) In summary, the EPR analysis of Mn depletion in single PSII microcrystals established the standard conditions used in the subsequently described crystallographic investigation, namely Mn depletion of $C_{12}E_8$ dPSIIcc crystals by treatment with 50 mM $NH_2OH$/EDTA.

## Comparison between the structures of the $Mn_4CaO_5$-cluster depleted dPSIIcc and intact dPSIIcc

Data was collected on a single crystal of 100*80*60 $\mu m^3$ that was previously treated with 50 mM $NH_2OH$/EDTA and integrated to 2.55 Å resolution structure following the I/sigma(I) > 2 convention. Additional resolution shells extending to 2.2 Å were included in refinement. The structure was build based on the WOC-bound structure at 2.44 Å resolution previously obtained under the same crystallization conditions (PDB entry 4PJ0 [*Hellmich et al., 2014*]). Both the 4PJ0 structure and the 1.9 Å resolution structure (PDB entry 3WU2 [*Umena et al., 2011*]) were used for structural comparison. The overall structures are highly similar and Cα atoms could be superimposed to 4PJ0 with 0.209/ 0.225 Å rmsd (for the two PSII monomers) and 0.218/0.191 Å (for the two D1 subunits), and to 3WU2 with 0.333/0.257 Å (monomers) and 0.246/0.242 Å (D1). These differences are however consistent with Cα rmsd difference between the two reference structures and may reflect experimental differences such as buffer composition and pH, crystal packing and resolution. 700 water molecules were placed, representing a quarter of those observed at higher resolution in line with expectations (*Weichenberger et al., 2015*). Waters are predominantly located within the membrane extrinsic subunits and in channels and cavities leading up to the cluster. They occupy similar positions compared to the high resolution structure (Figure 4B; overlays are shown in *Figure 4—figure supplement 1A and B*); but in the apo-PSII structure we did not observe water molecules in place of the W1-4 oxoligands otherwise coordinating Mn4 and Ca.

Removal of the $Mn_4CaO_5$-cluster did not result in any discernible movement of subunits or domains, neither at the PSII donor side nor in the region of the membrane spanning helices or at the

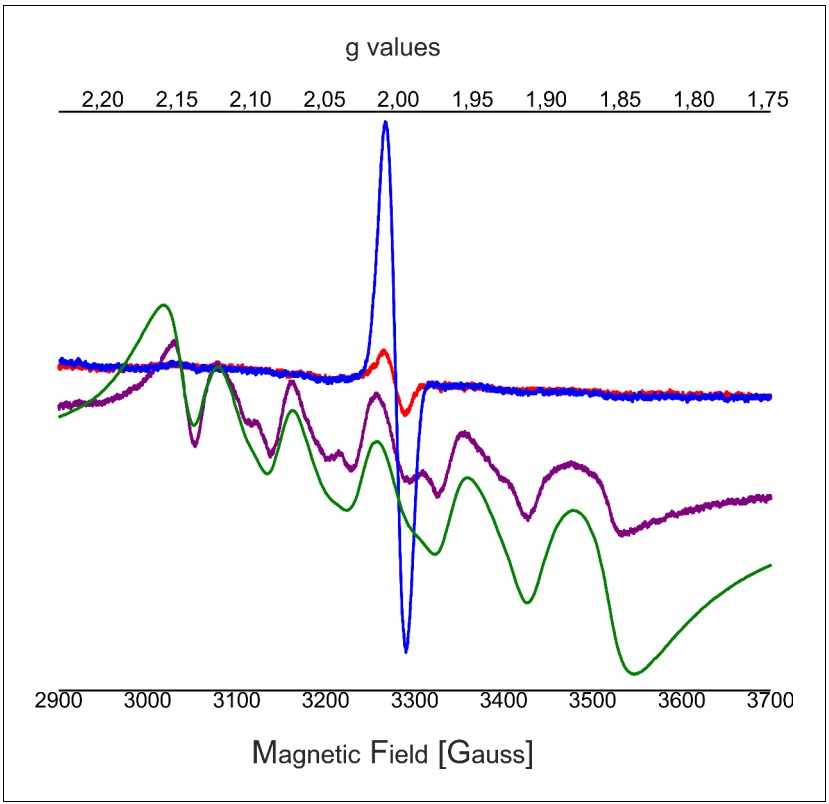

**Figure 2.** Influence of EDTA added to NH₂OH treated C₁₂E₈ dPSIIcc solutions. EPR spectra of untreated dPSIIcc samples (blue trace), 50 mM NH₂OH treated dPSIIcc (purple trace), 50 mM NH₂OH/EDTA treated dPSIIcc (red trace) and 1 mM dissolved MnCl₂ in dPSIIcc (green trace). Dimeric PSIIcc solutions were incubated in the presence of 50 mM NH₂OH (purple trace) for 10–15 min at RT in dark. The sample was further washed with 50 mM EDTA (red trace). Samples were measured at 20 K (microwave frequency, 9.22 GHz; field modulation amplitude, 32 G at 100 kHz; microwave power, 1 mW).

The following figure supplements are available for figure 2:

**Figure supplement 1.** EPR spectra after addition of NH₂OH using different concentrations.

**Figure supplement 2.** EPR spectra of added $Mn^{2+}$ with apo-WOC-PSIIcc using 10 mM MnCl₂ washed with buffers with CaCl₂.

---

acceptor side, as illustrated by an overlay of the $C_\alpha$ atoms in D1 (*Figure 3—figure supplement 1*). However, the extrinsic PsbU subunit in the monomer not involved in a crystal contact appeared destabilized (*Figure 3*), in line with lability of PsbU binding to PSIIcc in solution in the presence of NH₂OH/EDTA as discussed above. Structures of thermophilic PSII presented to date share the same $P2_12_12_1$ space group and crystal packing while dehydration (*Umena et al., 2011*) or detergent depletion (*Hellmich et al., 2014*) have resulted in different unit cell size. In both cases, crystal packing relies on contacts between PsbU and PsbV. The crystal contact between the extrinsic subunits PsbU and PsbV acts as a pivot between adjacent PSII dimers (*Figure 3A*, inset). Within PSII, PsbU acts as a lid which extends from its closest point (Tyr 103), 14 Å away from the WOC metal-cluster, to 50 Å away. In the crystal packing the two monomers differ as in one monomer PsbU interacts with PsbV' of a symmetry related PSII dimer in the crystal (locked monomer) while the other monomer (unlocked monomer) does not show this crystal contact between PsbU and PsbV'. While PsbU of both monomers is well resolved, the electron density map of PsbU in the unlocked monomer (which is not stabilized by a crystal contact) is noticeably perturbed in the present structure and apparently more mobile, as judged by atomic B-factors (*Figure 3A*). The increased disorder follows through the entire subunit in the one monomer and is translated into the WOC binding site by an interaction of

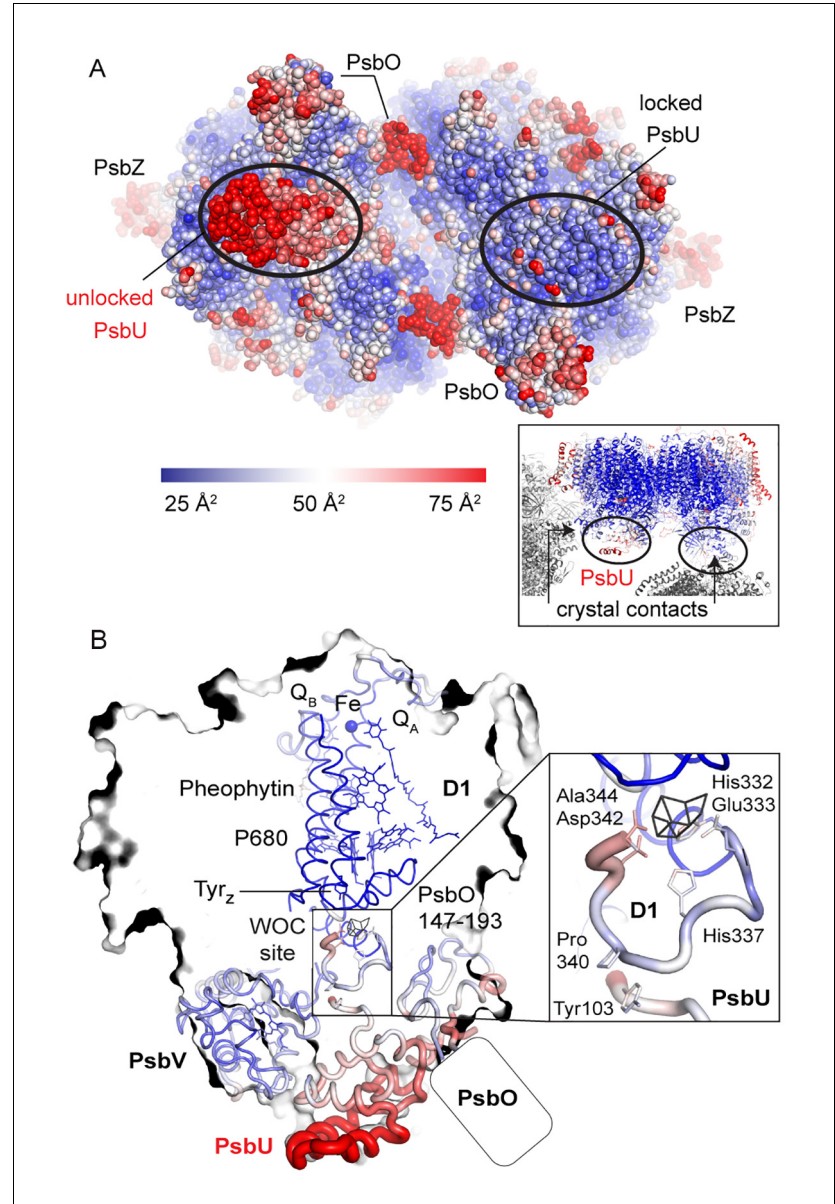

**Figure 3.** Comparison between the locked and unlocked monomer. (**A**) Locked (right) and flexible, unlocked monomer (left). Atomic B-factors are shown using the indicated color coding. The inset shows the crystal contacts acting on PsbU in the right monomer. (**B**) A slice through the unlocked PSII monomer showing D1 and extrinsic, manganese stabilizing subunits PsbU, PsbV, PsbO (the 147–193 loop and the position of the external β-barrel). The inset shows the WOC-binding site and C-terminal loops of both PsbU and D1 including, residues involved in their interaction (Tyr103 and Pro340) and residues involved in cluster binding in intact PSII.

The following figure supplements are available for figure 3:

**Figure supplement 1.** Overlay of $C_\alpha$ atoms (spheres) of D1 residues between the WOC site and the acceptor site (Fe, $Q_A$, $Q_B$).

**Figure supplement 2.** Increase in B-factors (blue, less flexible to red, more flexible) between PsbU and the WOC site in the unlocked (left) monomer, but much less so in the monomer locked by a crystal contact (right).

the C-terminal loop of PsbU at Tyr103 which stacks against Pro340 in the C-terminal loop of D1 (see *Figure 3B* inset; *Figure 3—figure supplement 2*). This D1 loop connects His337 (μ-oxo-bridge binding), His332, Glu333, Asp342 and the Ala344 carboxyl-terminus which are all ligands to Mn1-4 and Ca. The aromatic structure of Tyr103 in the PsbU protein of red algal PSII has been shown to be important for the optimal function of the WOC (*Okumura et al., 2007*). Interestingly, in the recent cryo-EM structure of spinach PSII, a loop from PsbP takes the place of the PsbU C-terminus (*Wei et al., 2016*). At the WOC binding site, all cluster-ligating side chains, with the exception of Asp170 (located at the opposite side of the WOC site), are more disordered in the unlocked monomer than in the locked monomer even though the cluster is absent in both. From the B-factor analysis (*Figure 3A*) we suggest that accessibility and stability of the WOC of PSII treated with reducing and chelating agents may be influenced by a crystal contact involving PsbU. In the locked monomer the WOC is well protected by the intact PsbU subunit (see crystal contact discussed above in *Figure 3A*, inset, right) whereas in the unlocked monomer the WOC is easier accessible due to the more flexible, perturbed PsbU (no crystal contact in *Figure 3A*, inset, left). Thus, crystal packing may have a direct effect on the WOC site. Also in the 2.44 Å resolution structure of the dPSIIcc with fully assembled WOC (*Hellmich et al., 2014*), both monomers differ slightly in the PsbU structure (*Figure 3—figure supplement 2E*), albeit more subtle than the two monomers of apo-PSII. Similarly, the XFEL data from *Young et al. (2016)*, showed a small difference between the two monomers. These results differ from the conclusions drawn by *Tanaka et al. (2017)* based on their high-resolution structures of PSII, where a difference between the WOC in the two monomers was also observed; they interpreted that the crystal packing most likely does not have an effect, and the structural differences between the two monomers were likely originated from different S-states of the WOC. In the following section, we discuss the structure of the better ordered apo-PSII of the locked monomer (for structures of the unlocked monomer, see Supplementary).

The structure of PSII treated with 50 mM $NH_2OH$/EDTA in the presence of 100 mM $(NH_4)_2SO_4$, 100 mM Tris-HCl, pH 7.5, and 35% PEG 5000 MME, but in the absence of calcium, magnesium or manganese ions displays a clear lack of defined electron density at the site of the WOC (*Figure 4B and C*, *Figure 4—figure supplement 1*; see also *Figure 4—figure supplements 2* and *3* for stereo views). Cluster-coordinating protein side chains nevertheless are found in positions very similar to the native structure. Rotation of side chain carboxyl and imidazol moieties are observed, but their new rotameric positions are only partially defined. In the WOC-bound dPSIIcc (*Umena et al., 2011*), the cluster is coordinated by six carboxylates - most of which act as ligand to two metal ions – and a single histidine. Mn1 and Mn2 are bound by three protein ligands, the central Mn3 by two protein ligands, and Mn4 and Ca by two protein and two water ligands each. In the apo-WOC structure, those protein side chains largely retain their orientation, within the limits of accuracy of the present 2.55 Å resolution structure (*Figure 5*). Only Glu354, coordinating Mn2 and Mn3 and contributed by CP43, is no longer defined in the electron density and presumably assumes a number of alternative positions.

## The TyrZ—His moiety neighboring the active-site metal-cluster

The electron transfer (ET) from a redox-active tyrosine residue (D1-Tyr161), denoted as TyrZ, to the chlorophyll cation radical, $P680^+$, which was created by primary charge separation, is coupled to movement of a proton to a neighboring histidine residue (D1-His190). This proton-coupled ET step has been found to be strongly slowed down upon Mn depletion and its rate becomes pH dependent (*Styring et al., 2012*). One conceivable explanation is breakage of the TyrZ-His190 hydrogen bond upon Mn depletion. A strong hydrogen bond of 2.4 Å length (*Umena et al., 2011*) connects the redox active TyrZ and a histidine in D1 in PSII with assembled metal-cluster, whereas for the apo-PSII we determine a distance of 2.8 Å (and of 2.6 Å in the second, unlocked PSII monomer). Yet this observation does not represent evidence for a real change in the TyrZ-O—N-His190 distance, because of the limited resolution of the apo-PSII structure. In conclusion, the TyrZ-O-H—N-His190 H-bond clearly is maintained in the apo-PSII, at least at the proton activity present in the crystals (pH around 7.0–7.5). Structural models at higher resolution are required to answer the question whether the H-bond becomes weakened by removal of the metal-cluster.



**Figure 4.** Comparison of intact WOC site and Mn-depleted site. (**A**) Coordination of the intact WOC by side chains from the D1 subunit (cream) and CP43 (green) and four water molecules (W1–W4). Arg357(CP43) and His 337(D1) form hydrogen bonds to WOC oxygen atoms (***Umena et al., 2011***). (**B**) Model of sidechains and electron density map (1 sigma 2Fo-Fc, grey) of the apo-WOC and electron density (0.8–1.5 sigma 2Fo-Fc, blue) tentatively modeled as waters at the site of the WOC and surrounding water molecules (blue). Multiple contour lines indicate a stronger (more defined) peak in the electron density map (e.g. waters at Asp170), while fewer lines show a weaker signal (e.g. the water at the WOC O2 site). (**C**) Overlay of apo-WOC model (green) with two holo-PSII

*Figure 4 continued on next page*

*Figure 4 continued*

models at a similar (PDB: 4PJ0, [*Hellmich et al., 2014*], light grey) and higher resolution (PDB: 3WU2, [*Umena et al., 2011*], dark grey).

The following figure supplements are available for figure 4:

**Figure supplement 1.** Overlay of the WOC-coordinating side chains at both monomers of dPSIIcc.

**Figure supplement 2.** Stereo view of the WOC binding site as shown in *Figure 4* but with the 0.8–1.5 σ $2F_o$-$F_c$ electron density map (blue) shown at water positions (blue spheres) and chloride positions (emerald spheres) only.

**Figure supplement 3.** Stereo view comparing water positions at the WOC site in apo-PSII (green) with those in 3WU2 (purple).

## What is there in place of the protein-bound $Mn_4CaO_5$-cluster and how could it promote photo-activation?

In each monomer, two peaks in the electron density map were observed in the absence of the WOC and these have been tentatively assigned as water molecules. They are approximately located at the positions of the two oxygen atoms (O2 and O3) that bridge between Mn2 and Mn3 in the intact complex (*Figure 4B*, *Figure 4—figure supplement 2*, *Figure 5—figure supplement 1D*). (The electron densities also might relate to ammonium or sodium ions from the crystallization buffer or from remaining manganese or calcium ions at low occupancy; yet binding of these cations to the previous $O^{2-}$ positions is unlikely.) Inspection of a space-filling model of the emptied WOC site suggests that the gaps between Van-der-Waals spheres do not exceed 3 Å (*Figure 5—figure supplement 1*), suggesting that two water molecules could be already sufficient to replace the 10 atoms of the $Mn_4CaO_5$ core of the assembled metal-cluster. Although on a first glance surprising, this is understandable: the two water molecules as well as the ligand atoms of the formerly metal-binding residues cover clearly more space in the emptied WOC site because they are not interconnected by coordination bonds; the sum of their Van-der-Waals radii (2 x $R_O$ equals about 3.1 Å) exceeds the metal-ligand bonding distance significantly (typical $Mn^{III/IV}$—O bond length of 1.9 Å). *Figure 5B* shows that after removal of the $Mn_4CaO_5$-cluster, several formerly coordinating carboxylate side chains are at H-bonding distances to each other ($\leq$3 Å; His332—Glu189—Asp342; (His337)—Asp342—OCO/Ala344), implying that one carboxylate of each H-bond pair is protonated. Since histidine residues in H-bonding distance to a carboxylate sidechain are most likely protonated, the protonation-state pattern indicated in *Figure 5B* is plausible. Assuming that in PSII with fully assembled metal-cluster as well as in the apo-PSII the overall net charge of the metal-cluster site is similar, presumably 2–3 further carboxylate residues (Glu/Asp) will be protonated in the apo-PSII. In conclusion, removal of the five metal ions does not create spatial gaps that would need to be filled by rearrangement of neighboring protein groups. Instead the arrangement of the metal-coordinating residues with and without the metal-cluster is largely the same. Consequently, the apo-PSII, which likely represents the starting state for metal-cluster (re-)assembly, is characterized by pre-formed manganese binding sites, set up to facilitate correct assembly of the $Mn_4CaO_5$-cluster. This chelate effect may be especially strong for Mn1 and Mn2 because there are three coordinating protein groups (versus two for Mn3 and Mn4). All in all, the above structural information points towards an intriguing mode of incorporation of Mn1 and Mn2 into the PSII apo-protein. Namely, deprotonation of the four H-bonding pairs allows for charge-compensated coordination of two Mn cations to the respective residues without any major movement of the coordinating imidazole and carboxylate sidechains.

## Possible intermediates of disassembly and photo-activation

After manganese depletion of the dPSIIcc crystals, we achieved partial reassembly of the metal cluster (see also Supplementary). The corresponding crystallographic data at 4.5 Å resolution revealed electron density centered between the Mn1 and Mn2 binding site (*Figure 6A* and *Figure 6—figure supplement 1A*), compatible with manganese binding at these two sites.

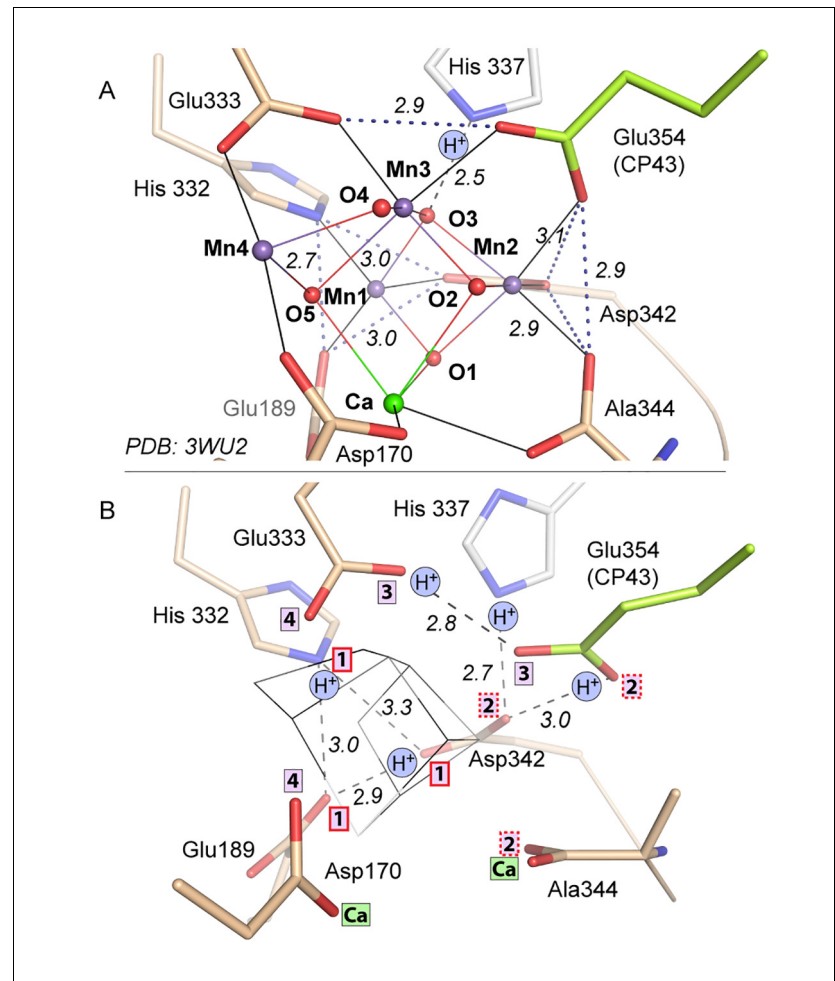

**Figure 5.** Comparison of intact WOC site and Mn-depleted site. **(A)** Intact WOC: Distances (Å) between O/N atoms of residues coordinated to the same metal ion. **(B)** Apo-WOC: Distances (Å) between O/N atoms which are likely connected by H-bonds. In **B**, also estimates for likely protonation states are shown. Boxed numbers indicate to which metal ion (Mn1 to Mn4; Ca) the respective O/N binds in the intact WOC. The His337 is H-bonded to a $\mu_3$-oxo bridge between Mn1, Mn2, and Mn3. Positions of Glu354 and Ala344 have higher uncertainty or mobility (see also *Figures 1C* and *4C*).

The following figure supplement is available for figure 5:

**Figure supplement 1.** Packing of water molecules in apo-PSII and µ-oxo-bridges in intact PSII.

We also conducted partial disassembly experiments (with lower NH$_2$OH concentration) in the crystals and the obtained electron density at 4.0 Å resolution is centered around the Mn1 and Mn2 site (*Figure 6B* and *Figure 6—figure supplement 1B*), similar to the situation observed for the partial reassembly experiment. Due to the low resolution, it remains uncertain whether a single Mn ion binds between the Mn1 and the Mn2 site or whether the Mn1 and Mn2 sites are both occupied by one manganese ion each. Taking into account the pre-formed Mn1 and Mn2 binding sites detected in the apo-PSII, the latter option is more plausible. From these low-resolution crystallographic experiments alone, the precise position, ligation, or occupancy of individual atoms could not be reliably determined. Previous experimental findings supporting a bi-manganese unit as an intermediate in the cluster assembly have been presented (see [*Bao and Burnap, 2016*] for review). Investigations involving X-ray absorption spectroscopy (XAS) support a di-µ-oxo bridged pair of Mn(III) ions as an intermediate in the heat-induced disassembly (*Pospíšil et al., 2003*), or as an intermediate state during photo-activation (*Barra et al., 2006*). The joint evidence from these earlier investigations and

**Figure 6.** Partially (dis)assembled WOC. (**A**) Difference electron density obtained for the partially reconstituted and (**B**) partially disassembled WOC. Grey spheres indicate the position of manganese atoms in 3WU2 when D1 Cα atoms are superimposed. Green mesh indicates 4.5 σ $F_o$-$F_o$(apo) maps at 4.5 Å (**A**) and 4 Å (**B**) resolution. (**C**) Possible Model describing the initial reassembly of the WOC based on the structural data for the partially reconstituted WOC. The scheme represents the binding and the light driven relocation/oxidation of the first two Mn ions into the apo-PSII. The first $Mn^{2+}$ (light orange) moves to the Mn4 position of the apo-PSII (green) and binds at this high affinity site (1). After light driven oxidation with low quantum efficiency a rearrangement takes place where the oxidized $Mn^{3+}$ (orange) migrates from the Mn4 position to the Mn1 position (2). The same rearrangement also applies to the second $Mn^{2+}$ which binds at the Mn4 position and then relocates as the oxidized $Mn^{3+}$ to the Mn2 position (3, 4). Both $Mn^{3+}$ ions form a stable Mn1-(μ-O)$_2$-Mn2 assembly intermediate (4), as shown in **A**. Binding of the two remaining Mn ions and the Ca ion (blue) into the apo-WOC has not been resolved yet (5). Please note that our data do not exclude that $Ca^{2+}$ could also bind in an earlier step, e.g. in step 3, 4 or 5.

The following figure supplement is available for figure 6:

**Figure supplement 1.** Reconstitution and disassembly of the WOC.

our structural data render it plausible that a partially (dis)assembled state of the metal-cluster is formed where the Mn1 and Mn2 site are occupied and interconnected by two bridging oxides (Mn1-(μ-O)$_2$-Mn2) in the same way as they are in the fully assembled $Mn_4CaO_5$-cluster.

Several earlier studies arrived at different conclusion whether $Ca^{2+}$ is required for proper assembly of the $Mn_4CaO_5$ cluster or not and at what stage of the photo-activation it is involved. The current consensus is that $Ca^{2+}$ is absolutely required for efficient photoassembly and it seems to affect the first photoactivation intermediate (*Bao and Burnap, 2016*). In our data we do not see evidence for significant occupancy of the calcium site in the partially assembled metal-cluster. This could be due to several reasons. In previous studies, a high $Ca^{2+}$ to $Mn^{2+}$ ratio was found to be important for the highest yield of photoactivation with an optimum ratio between 500:1 and 20:1. However, under our buffer conditions it is not possible to increase the calcium concentration (see Materials and methods). It could well be that due to the considerably lower concentration of calcium used we

populated an early intermediate state that only includes Mn. On the other hand, one must be aware that at the limited resolution of 4.5 Å achieved for the data of the partially reconstituted PSII it is difficult to locate or decide about the occupancy of calcium (for example it was not possible to locate Ca unambiguously in the initial structural data of intact PSII at 3.8 Å resolution (*Zouni et al., 2001*). For that reason, we cannot exclude that $Ca^{2+}$ is present in the observed assembly intermediate. It should also be noted that previous studies observed the formation of improper Mn clusters with >5 Mn in the absence of $Ca^{2+}$ (*Chen et al., 1995*) but we do not see any indication of a high number of Mn present in the WOC site, indicating that these clusters are not formed in dPSIIcc crystals under our conditions.

Recently, strong evidence based on pulsed EPR spectroscopy has been presented that a first step of the photo-activation process is the high-affinity binding of a $Mn^{2+}$ ion occurring already in the dark (before onset of illumination) by coordination to two specific axial ligands, Asp170 and Glu333, and thus at the Mn4 site (*Asada and Mino, 2015*), in line with earlier findings reviewed in (*Bao and Burnap, 2016*). Only Mn4 is coordinated by two carboxylates in trans position and we now find that this distinctive arrangement of Asp170 and Glu333 is maintained upon disassembly of the metal complex. This pre-organized binding site with two axial carboxylate ligands and room for four water ligands in the equatorial plane likely facilitates high-affinity binding of $Mn^{2+}$ ions in an octahedral geometry. As suggested by (*Bao and Burnap, 2016*), Mn may initially bind at the Mn4 site, before it relocates to the Mn1 or Mn2 site. The latter is in line with our observations. Here, we propose a possible model of the initial WOC-reassembly (*Figure 6C*) based on our partial assembly data. The movement of the Mn ion initially bound to the Mn4 site towards the Mn1 site could relate to the dark rearrangement step in the two-quantum model of photo-activation (*Cheniae and Martin, 1971*; *Dasgupta et al., 2008*; *Becker et al., 2011*; *Bao and Burnap, 2016*).

## Comparison with other metalloenzymes

The structure of the metal-cluster site is largely identical when comparing fully functional PSII and apo-PSII, in remarkable contrast to many other metalloenzymes. For example, a crystal structure in which the insertion of the catalytic [Mo-7Fe-9S-C] cluster (*Spatzal et al., 2011*) of MoFe-containing nitrogenase was interrupted differs remarkably from the holo-protein and shows substantial conformational re-modeling in one of its three subunits whereby a positively charged funnel is created along which the cofactor may be inserted (*Schmid et al., 2002*). Furthermore, several side and main chain conformations around the cluster cavity vary between cofactor-free and –bound nitrogenase (*Schmid et al., 2002*). A similar situation was reported for FeFe-hydrogenase, where the insertion of the 2Fe sub-cluster to form the H cluster, where the 2Fe sub-cluster is linked to a [4Fe4S]-cluster (*Peters et al., 1998*; *Nicolet et al., 1999*), is helped by conformational changes including the formation of a positively charged tunnel from the surface to the active site (*Mulder et al., 2010*). Although dPSIIcc also contains a complex heterometal-cluster, we observed only minor conformational differences between WOC-containing and -free form. As the WOC-free state was generated in the crystal, the absence of conformational changes in PSII may be due to crystal lattice forces, which can not only restrain the dissociation of subunits (PsbU, PsbV and PsbO) but also larger conformational changes near the WOC-binding site. However, local conformational changes in the cavity of the WOC are not restrained by the crystal packing as the residues are directed into a solvent filled cavity. Yet, the side chains do not move.

The dissimilarity with respect to the assembly of other complex metal sites likely roots in the fact that the WOC is a pentanuclear metal complex with a unique assembly pathway. In contrast to other complex metalloenzymes, whose metal-clusters are assembled on accessory scaffold proteins before being inserted into the target protein, the WOC is assembled in situ, in a multi-step process involving the oxidation of Mn ions at the metal binding site by the tyrosine radical (TyrZ•) formed under illumination (*Cheniae and Martin, 1971*; *Dasgupta et al., 2008*; *Becker et al., 2011*; *Bao and Burnap, 2016*). We now find that in the apo-PSII, the ligand shell of the $Mn_4$Ca-oxo core is pre-organized in a configuration that is surprisingly close to that of the fully assembled metal complex. This almost perfectly pre-organized ligand shell likely facilitates kinetically competent and error-free formation of the metal complex upon light-driven oxidation of Mn ions. Interestingly, although the WOC is unique regarding structural complexity of the $Mn_4$Ca metal site and light-driven Mn oxidation during assembly of the metal complex, binuclear Fe or Mn sites found in other proteins not only

share similar coordinating residues and bridging ligands, but may also share a similar mode of oxidative self-assembly (*Griese et al., 2014*).

### Concluding remarks

We obtained the PSII structure for crystals completely depleted of the $Mn_4CaO_5$-cluster. This structure serves as a basis to understand the mechanism of WOC assembly/disassembly in PSII. Our new apo-PSII structure showed a pre-organized ligand shell, including protein sidechains and water molecules assisting the stepwise auto-assembly of the active WOC by offering a structuring template, directing metals and waters to places where they can be photo-activated/oxidized with minimal restructuring. Moreover, we observed metal binding at the Mn1 and Mn2 site in the partially disassembled and partially reassembled metal-cluster and tentatively assign this assembly intermediate to the di-μ-oxo bridged $Mn(III)_2$ cluster (Mn1-$(μ$-$O)_2$-Mn2) previously detected by X-ray absorption spectroscopy (XAS) (*Pospíšil et al., 2003*; *Barra et al., 2006*). Our observations are thus in line with the XAS data in PSII solutions and compatible with the two-quantum model of photo-activation (*Cheniae and Martin, 1971*; *Dasgupta et al., 2008*; *Becker et al., 2011*; *Bao and Burnap, 2016*). Many facets of the light-driven formation of the $Mn_4CaO_5$-cluster remain enigmatic, yet the herein described approach of partial (and full) disassembly and partial reassembly in dPSIIcc crystals paves the road for detailed structural characterization of assembly intermediates at high resolution. Further and detailed reconstitution experiments involving different buffer solutions under variations of the $Ca^{2+}$ / $Mn^{2+}$ ratio under different light conditions on apo-PSII microcrystals utilizing membrane inlet mass spectroscopy are planned in the near future to determine optimum conditions for full reconstitution of the $Mn_4CaO_5$ cluster in the apo-PSII crystals.

## Materials and methods

### Protein purification

The cell cultivation and preparation of thylakoid membranes of *T. elongatus* were performed as described in (*Kern et al., 2005*; *Hellmich et al., 2014*). Thylakoid membranes were solubilized with 1.2% (w/w) $C_{12}E_8$ in 0.02 M MES/NaOH, pH 6.0, 0.02 M $CaCl_2$, 0.01 M $MgCl_2$, 0.5 M Betaine (MCMB) at a chlorophyll (Chl.) concentration of 1.7 mM for 5 min at room temperature (RT) in darkness. The incubation was stopped by adding buffer containing 0.02 M MES/NaOH, pH 6.0, 0.01 M $CaCl_2$, 0.5 M Betaine and 0.013% $C_{12}E_8$ (MCB) in ratio 1:1. The suspension was centrifuged at 14000 rpm for 30 min at 4°C. The supernatant defined as protein extract was collected for the subsequent purification. The purification of the extract was implemented according to (*Hellmich et al., 2014*). PSII extract was loaded onto a Toyopearl DEAE 650 s (Tosoh Bioscience) weak anion exchange chromatographic column in MCB buffer, and eluted with a linear gradient from 0 to 0.1 M $MgSO_4$. The last step of the purification was to use a MonoQ 10/100 GL (GE Healthcare) column in MCB buffer, the column was subjected to a linear gradient from 0 to 1 M NaCl. The $C_{12}E_8$ dPSIIcc obtained from the main peak fractions were finally concentrated using Vivaspin 20 ultrafiltration membranes with 100 kDa cutoff until a final Chl concentration of 2 mM was reached. All the purification steps were carried out at 4°C. The PSII core complexes were stored in liquid nitrogen.

### Oxygen-evolution measurements

The oxygen-evolution activities of $C_{12}E_8$ dPSIIcc samples were measured on a Clark-type electrode under continuous light illumination in the presence of 1 mM 2.5-Dichloro-1,4-benzoquinone (DCBQ) as the electron acceptor at 2.7 μg Chl per 1 ml in MCMB buffer, pH 6.5. For oxygen inactivation measurements, dPSIIcc samples were incubated in darkness at 25°C for 30 s with 0 mM to 2 mM $NH_2OH$ prior to illumination.

### Sodium dodecylsulfate polyacrylamide gel electrophoresis (SDS-PAGE)

To detect the lager PSII subunits, SDS-PAGE gel at 12% was used. 20 μg protein of PSIIcc was suspended in buffer with MCB for $C_{12}E_8$ dPSIIcc, and denatured with the loading buffer in 4:1 ration at 60°C for 10–15 min. Afterwards, samples were electrophoresed in an SDS-PAGE with a 12% gradient gel according to *Laemmli (1970)*.

## Blue native PAGE (BN-PAGE)

BN-PAGE was carried out using Serva Gel N 3–12 (Vertical Native Gel 3–12%). PSII core complexes at 2 mM were mixed in ration 1:1 with BN-PAGE sample buffer (Serva) before loading. Electrophoresis was run in the cathode buffer A containing 50 mM Tricine, 7.5 mM Imidazol, 0.02% Coomassie G250, pH 7.0. Anode buffer contained 25 mM Imidazol, pH 7.0. After 2/3 of the run, the cathode buffer is replaced by cathode buffer A with lower Coomassie concentration (0.002%) in order to gain clearer visualization of blue bands.

## Size exclusion chromatography

To purify the PSII apo protein from any dissociated subunits after $NH_2OH$/EDTA treatment $C_{12}E_8$ dPSIIcc samples were run on a Superose six column (GE Healthcare). MCB buffer without $CaCl_2$, added with 0.05 M $MgSO_4$ was used for equilibration at a flow rate of 0.3 ml/min. PSII fractions containing dimer or/and monomer were collected after elution from the column and were concentrated by ultrafiltration using 100 kDa cutoff. UV-absorption was detected at 280 nm.

## MALDI-TOF-MS

Protein masses were analysed by matrix-assisted laser desorption ionization-time of flight mass spectrometry (MALDI-TOF-MS) using an Ultraflex-II TOF/TOF instrument (Bruker Daltonics, Bremen, Germany) equipped with a 200 Hz solid-state Smart beam laser. The mass spectrometer was operated in the positive linear mode. MS spectra were acquired over an m/z range of 3000–30,000 and data was analyzed using FlexAnalysis 2.4. software provided with the instrument.

Sinapinic acid was used as the matrix (saturated solution in acetonitrile/0.1% trifluoroacetic acid 1:2) and samples were spotted undiluted using the dried-droplet technique. If necessary, samples were diluted in $TA_{33}$ (33% acetonitrile/0.1% trifluoroacetic acid in water).

## Protein crystallization

According to the crystallization protocol described in *Hellmich et al. (2014)*, $C_{12}E_8$ dPSIIcc crystals were obtained using *microbatch* 96 Well IMP@CT Plates (Greiner Bio-One, Frickenhausen, Germany). The dimeric $C_{12}E_8$ PSIIcc at 2 mM Chl. solubilized in buffer containing 0.02 M MES/NaOH, pH 6.0, 0.01 M $CaCl_2$, 0.5 M Betaine and 0.013% $C_{12}E_8$ (MCB) buffer was mixed in 1:1 ratio with the crystallization solution which contained 0.1 M TRIS/HCl pH 7.5, 0.1 M $(NH_4)_2SO_4$ (TN buffer) 14–18% (w/v) polyethylene glycol 5000 monomethyl (PEG 5000 MME). Crystals grew at 20°C and appeared after 2–3 day having a size between 50–250 μm.

## Complete and partial $Mn_4CaO_5$-cluster depletion on dPSIIcc crystals

For complete depletion, the dPSIIcc crystals were incubated in TN buffer, 15–20% PEG 5000 MME at RT for 60 min in darkness. After incubation, initial buffer was exchanged by TN buffer with 20% PEG 5000 MME, 50 mM $NH_2OH$, 50 mM EDTA for 30 min to release Mn. The $Mn_4CaO_5$-cluster depleted dPSIIcc crystals were washed three times in TN buffer, 20% PEG 5000 MME without $NH_2OH$ and EDTA. From this step, the PEG concentration was stepwise increased (5% steps, 15 min) following the dehydration protocol described in *Hellmich et al. (2014)*. The final TN buffer contained 35–40% (w/v) PEG 5000 MME. For partial depletion, various concentrations of $NH_2OH$/EDTA from 30 mM to 40 mM were used and the incubation time in $NH_2OH$/EDTA containing buffer was reduced down to 10 min.

## Partial reconstitution of the $Mn_4CaO_5$-cluster

The apo-PSIIcc crystals (treated with 50 mM $NH_2OH$ and EDTA as described in previous section) were used for the partial reconstitution. The crystals were washed in a $NH_2OH$ and EDTA free buffer (TN buffer, 20% PEG 5000 MME) for three times and were then transferred in a buffer containing 10 mM $MnCl_2$, 10 mM $NaHCO_3$, 5 mM $CaCl_2$, 0.1 M TRIS/HCl pH 7.5, 0.1 M $(NH_4)_2SO_4$, 20% PEG 5000 MME (reconstitution buffer) under dim green light (3 min, 2.6 μE/m²/s). To avoid the precipitation of $CaSO_4$ under the buffer conditions, the maximal concentration of $CaCl_2$ was limited to 5 mM. Following 15–20 min of dark incubation in the reconstitution buffer at RT, the crystals were transferred into the TN buffer with 20% PEG 5000 MME and washed three times in order to remove excess Mn. In each buffer exchange step during the crystal treatment and dehydration procedure (increase of

PEG 5000 MME in 5% steps to reach a final concentration of 35%, 15 min incubation time per step), the crystals were illuminated intermittently with green light of the microscope (2.6 µE/m$^2$/s), yielding in a total illumination time of 20 min over the entire procedure. Using an extinction coefficient of 5.8 mM$^{-1}$cm$^{-1}$ at 532 nm, a Chl concentration of 59 mM in the crystals and a crystal thickness of 50 µm we estimate a total of 15 absorbed photons per reaction center in the crystals. No additional electron acceptors were added but previous studies on similar samples revealed that a small quinone pool is present in the crystals that allows transfer of 3–6 electrons (*Krivanek et al., 2007*; *Young et al., 2016*).

## EPR measurements

Prior to measurements, both dPSIIcc solutions and microcrystals were incubated in MCB buffer using higher $C_{12}E_8$ concentration at 0.02% and various $NH_2OH$ concentrations of 1 mM, 5 mM, 10 mM, 20 mM, 30 mM, 50 mM, 100 mM were added to reduce Mn ions. The incubation was performed at RT for 10–15 min in darkness. The final Chl concentration was approximately 1.5 mM for solutions and ~0.3–0.4 mM for microcrystals.

Low temperature X-band EPR spectra were recorded using a Varian E109 EPR spectrometer equipped with a model 102 microwave bridge. Sample temperature was maintained at 20 K using an Air Products LTR liquid helium cryostat. Spectrometer conditions were as follows: microwave frequency, 9.22 GHz; field modulation amplitude, 32 G at 100 kHz; microwave power, 1 milliwatt. After measurement of each sample in the respective $NH_2OH$ concentration, the samples were then brought to 100 mM $NH_2OH$ concentration and measured again. The measurements after 100 mM $NH_2OH$ treatment were used to quantify the total Mn (II) content in the samples and were further used for normalization.

For Mn-reassembly measurements, a final concentration of 10 mM $MnCl_2$ was added to the Mn-depleted PSIIcc solution samples. The samples were incubated in the dark for 10 min, followed by washing the samples three times with buffer (20 mM MES pH 6.0, 0.5 M Betaine and 0.02% $C_{12}E_8$) containing 10 mM $CaCl_2$ and without $CaCl_2$.

## X-ray diffraction data collection and analysis

Experiments were performed at beamline 14.1 operated by the Helmholtz-Zentrum Berlin (HZB) at the BESSY II electron storage ring (Berlin-Adlershof, Germany) (*Mueller et al., 2012*). Data were collected from a 100*80*60 µm$^3$ crystal using a 50 µm diameter beam at 0.91841 Å X-ray wavelength. Data were integrated to 2.2 Å maximum resolution with XDS (*Kabsch, 2010*) and XDSAPP (*Krug et al., 2012*). At 2.55 Å resolution, the signal reached its limit (1.85) by mean I/sigma(I) > 2 convention. While paired refinement (*Karplus and Diederichs, 2012*) including progressive resolution shells did not give a clear indication of useful resolution, the extended resolution allowed the placement of additional water molecules without detrimental effects on model geometry and only modest increase in R/Rfree against the 2.55 Å resolution data. We thus call this a 2.55 Å resolution structure for comparability but include the additional resolution shells to 2.2 Å resolution (CC1/2 = 13%) in refinement and data submitted to the PDB. *Table 1* shows statistics for both data cutoffs.

Refinement and model building followed that of *Hellmich et al. (2014)* (PDB entry 4PJ0) and both structures were superimposable with 0.253 Å rmsd. The model was fitted to the electron density using iterative cycles of hand building in COOT (*Emsley et al., 2010*) and automated refinement in phenix.refine (*Afonine et al., 2012*). Distances from metals (in chlorophyll, heme and the none-heme iron) to coordinating residues were restrained to dictionary values and those from the bicarbonate to the non-heme iron and Tyr244 (D2) and Glu244 (D1) were restrained to values from PDB entry 3WU2. The quality of the protein model was confirmed with MolProbity (*Chen et al., 2010*). 96% of backbone torsion angles were in favored regions of the Ramachandran plot, 0.3% were outliers and the remaining residues were in additionally allowed regions.

The final model was deposited in the protein data bank (PDB: 5MX2).

For the partial dis- and reassembly experiments, crystals having the sizes between 50 and 100 µm length were tested. For partial disassembly, data were collected at the BESSY using the same equipment as described above. For reconstitution, data were collected at the Advanced Light Source (ALS), beamline 5.0.2. using a 20 µm diameter beam.

**Table 1.** Data collection and refinement statistics.

| | 2.55 Å data | Extended 2.2 Å data |
| --- | --- | --- |
| Data collection | | |
| Wavelength | 0.9184 | 0.9184 |
| Resolution range | 49.65–2.55 (2.641–2.55) | 49.65–2.197 (2.276–2.197) |
| Space group | $P\,2_1\,2_1\,2_1$ | $P\,2_1\,2_1\,2_1$ |
| Unit cell (Å, °) | 116.33, 219.62, 304.04 90, 90, 90 | 116.33, 219.62, 304.04 90, 90, 90 |
| Total reflections | 3348095 (259200) | 4094682 (131257) |
| Unique reflections | 252839 (25073) | 364603 (25404) |
| Multiplicity | 13.2 (10.3) | 11.2 (5.2) |
| Completeness (%) | 1.00 (1.00) | 0.93 (0.65) |
| Mean I/sigma(I) | 13.10 (1.85) | 9.34 (0.44) |
| Wilson B-factor | 36.00 | 37.11 |
| R-merge | 0.208 (1.211) | 0.2500 (3.270) |
| R-meas | 0.2163 (1.275) | 0.2621 (3.629) |
| $CC_{1/2}$ | 0.996 (0.752) | 0.993 (0.1320) |
| Refinement | | |
| R-work | 0.1912 (0.2890) | 0.2115 (0.3998) |
| R-free | 0.2414 (0.3250) | 0.2596 (0.4298) |
| Number of non-H atoms | | 50407 |
| macromolecules | | 41206 |
| ligands | | 8498 |
| Protein residues | | 5428 |
| RMS(bonds) | | 0.012 |
| RMS(angles) | | 1.17 |
| Ramachandran favored (%) | | 96 |
| Ramachandran outliers (%) | | 0.33 |
| Rotamer outliers (%) | | 1 |
| Clashscore | | 7.99 |
| Average B-factor | | 36.05 |
| macromolecules | | 36.76 |
| ligands | | 33.11 |
| solvent | | 30.15 |

Values in parentheses refer to the highest resolution shell.

R-meas: Redundancy-independent merging R-factor (**Diederichs and Karplus, 1997**).

$CC_{1/2}$: Pearson correlation coefficient of half data sets (**Karplus and Diederichs, 2012**).

## Acknowledgements

Crystal testing and data collection were performed at beamline 14.1 operated by the Helmholtz-Zentrum Berlin (HZB) at the BESSY II electron storage ring (Berlin-Adlershof, Germany) and BL5.0.2 of the Advanced Light Source at the Lawrence Berkeley National Laboratory, Berkeley CA, which is a DOE Office of Science User Facility under contract no. DE-AC02-05CH11231. We thank the support staff at BESSY II, ALS and at the beamline P11 at the light source PETRA III at DESY, a member of the Helmholtz Association. Thanks to Dr. Vittal Yachandra for discussions, to Franklin Fuller and Iris Young for testing crystals at ALS, to Dr. Mahdi Hejazi and Dr. Christoph Weise for the MALDI-TOF

measurements, to Tobias Werther (AG Dobbek) for his support at BESSY, to Dörte DiFiore, Ina Seuffert and Julia Wersig for technical assistance. This work was supported by Deutsche Forschungsgemeinschaft through the cluster of excellence 'Unifying Concepts in Catalysis' coordinated by the Technische Universität Berlin (Project E2/E3), Sonderforschungsbereich 1078 (Project A5) coordinated by the Freie Universität Berlin and the Human Frontiers Science Project Award No. RGP0063/2013 310 (AZ, JY, RH) and the Director, Office of Science, Office of Basic Energy Sciences, Division of Chemical Sciences, Geosciences, and Biosciences of the Department of Energy (DOE) under contract DE-AC02-05CH11231 (JY).

## Additional information

### Funding

| Funder | Grant reference number | Author |
|---|---|---|
| Deutsche Forschungsgemeinschaft | Unifying Concepts in Catalysis, Project E2/E3 | Miao Zhang<br>Holger Dau<br>Holger Dobbek<br>Athina Zouni |
| Deutsche Forschungsgemeinschaft | Sonderforschungsbereich 1078, Project A5 | Martin Bommer |
| National Institutes of Health | GM055302 | Ruchira Chatterjee<br>Jan Kern |
| Human Frontier Science Program | Project Award No. RGP0063/2013 310 | Rana Hussein<br>Junko Yano<br>Athina Zouni |
| Biosciences of the Department of Energy | Contact number: DE-AC02-05CH11231 | Junko Yano<br>Jan Kern |

The funders had no role in study design, data collection and interpretation, or the decision to submit the work for publication.

### Author contributions

MZ, Formal analysis, Methodology, Writing—original draft, Designed experiment and developed the crystallization protocol for Mn depletion and reconstitution, Tested, collected X-ray diffraction data, The analytical data were performed with help from RH; MB, Data curation, Formal analysis, Writing—original draft, Designed experiment, Tested, collected X-ray diffraction data, Build the structure and created the model art work under guidance of HDa and with input from MZ, AZ and HDo; RC, Formal analysis, Tested, collected X-ray diffraction data, Collected and analyzed EPR data; RH, Formal analysis; JY, Designed experiment; HDa, Writing—original draft, Designed experiment; JK, Designed the experiment, Tested, collected X-ray diffraction data; HDo, Data curation, Writing—original draft, Designed experiment; AZ, Conceptualization, Writing—original draft, Designed experiment

### Author ORCIDs

Miao Zhang, http://orcid.org/0000-0003-4022-7093
Holger Dobbek, http://orcid.org/0000-0002-4122-3898
Athina Zouni, http://orcid.org/0000-0003-0561-6990

## Additional files

### Supplementary files

• Supplementary file 1. Effect of $NH_2OH$ on inhibition of oxygen evolution rates in intact $C_{12}E_8$-dPSIIcc. The oxygen evolution was assayed at 25°C using different concentrations of $NH_2OH$ from 0 mM to 2 mM. *The rates are the averages of at least three repeated measurements. [†]The percentage of control rates are given in parentheses.

• Supplementary file 2. Completeness of PSII subunits after NH₂OH/EDTA treatment. ‡ n-Dodecyl ß-D-maltoside Masses from linear mode MALDI-TOF-MS and assigned to PSIIcc subunits. The experimental determined mass from the $C_{12}E_8$ dPSIIcc treated with 50 mM NH₂OH/EDTA is the average mass from spectra recorded from three independent sample preparations. n.d.: not detectable

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
