## [Decision Letter]

Thank you for submitting your article "Structural insights into the light-driven auto-assembly process of the water-oxidizing Mn_4_CaO_5_-cluster in photosystem II" for consideration by *eLife*. Your article has been reviewed by two peer reviewers, and the evaluation has been overseen by a Reviewing Editor and Philip Cole as the Senior Editor. The following individual involved in review of your submission has agreed to reveal his identity: Nathan Nelson (Reviewer #1).

The reviewers have discussed the reviews with one another and the Reviewing Editor has drafted this decision to help you prepare a revised submission.

Essential revisions:

The primary concerns the reviewers have are related to the reconstitution experiments. As presented, there are so few details in the paper that the reconstitution experiments could not be replicated independently, and without such details, the reviewers cannot fully assess the validity of the experiments. This issue must be addressed before the paper can be accepted.

Reviewer #1:

Since George Cheniae's early discoveries in the early 1970's, it is known that hydroxylamine causes reduction of Mn in the manganese cluster of PSII followed by loss of oxygen evolution. In cyanobacteria, the loss of the functional manganese cluster results in the loss of the three extrinsic subunits PsbU, PsbV and PsbO following by practical disassembly of the dimeric PSII complex. Reactivation of PSII involves denovo synthesis and assembly of subunits, incorporation of lipids and pigments and finally assembly of the manganese cluster. The stepwise process of photo-activation involves the incorporation of the Mn and Ca ions into the dimeric PSII core complex. First, a Mn(II) ion binds in the dark and is oxidized after absorption of a first light quantum forming an unstable Mn(III) intermediate. Following a light independent rearrangement, a second light quantum drives the oxidation of a second Mn(II) ion resulting in a next assembly intermediate. The incorporation of the other two missing Mn and Ca ions to complete the metal-cluster is not clearly resolved yet.

Recently "Quebane"-like manganese clusters were synthesized demonstrating the stability and possible self-assembly of such complex. (Barber 2017). However the precise arrangement and the interdependency with the coordinating ligands is not settled. The novelty of this work is that PSII crystals were used for stabilizing the environment of the manganese cluster such that it could be totally depleted without alteration not only in the protein clusters but also of the bound water molecules in its vicinity. This is a remarkable achievement. However the characteristic EPR TyrD● signal was significantly weaker than the signal of the untreated sample presumably because the TyrD● radical was reduced after addition of NH2OH. This alteration might be one of the obstacles that prevented a complete restoration of the oxygen evolution. Even though only partial reconstitution of the manganese cluster was achieved there are sufficient new observations that were achieve at high proficiency to merit publication of the manuscript. This manuscript also can set example for many other unrelated studies that report crystal structures without taking advantage of experiments that could be performed using the crystals.

Reviewer #2:

This manuscript by Zouni and coworkers describes a high-resolution crystal structure of Photosystem II (PSII) in which the Mn_4_CaO_5_ active site of water oxidation has been removed. Slightly lower resolution structures of partially disassembled and reassembled PSII are also presented. The most striking finding from this structural data is that the protein environment around the active site is basically unchanged in the absence and presence of the Mn cluster. Evidence is presented that in the apo reaction center, only a couple of water molecules fill the void and maintain the geometry of the ligand shell. Therefore, the protein is poised to efficiently assemble the Mn cluster with minimal conformational changes. This result will surprise and impress many in the photosynthesis and bioinorganic research communities.

All my concerns are with the reconstitution data.

The authors describe their reconstitution protocol in subsection “Reconstitution of the Mn_4_CaO_5_-cluster**”**. However, not enough detail is provided. Was light provided? What is the occupancy of the QB site? Were electron acceptors added? If no light was used, then only one Mn(II) would be expected to bind to the high affinity site. Only if light was provided is it possible for more than one Mn to be observed.

In addition, the MnCl_2_ and CaCl_2_ concentrations used (10 mM and 5 mM, respectively) are not compatible with efficient photo-assembly. In photo-assembly studies by Cheniae, Burnap, Dismukes, and Brudvig, Ca^2+^ concentration is always much higher than Mn^2+^ (as is the case in vivo). Maintaining a high [Ca^2+^]/[Mn^2+^] ration is critical for preventing improperly formed Mn clusters containing >5 Mn per PSII (see Chen, Cheniae 1995). The authors should show that their conditions promote a high yield of photo-assembly with ~4 Mn per PSII.

It is surprising and concerning that electron density from Ca^2+^ is not observed during reconstitution. In subsection “Possible intermediates of disassembly and photo-activation**“,** the authors state "It is disputed whether the calcium ion binds at an early or late photo-activation state" and cite a review by Bao and Burnap. This statement is a mischaracterization of the review and is at odds with the EPR results from Tyryshkin, Dismukes, and coworkers (2006) that Ca^2+^ is present near the first mononuclear Mn3+ intermediate of photo-assembly. The absence of Ca^2+^ makes the possibility of an improperly assembled cluster more likely (see previous paragraph).

[Editors' note: further revisions were requested prior to acceptance, as described below.]

Thank you for resubmitting your work entitled "Structural insights into the light-driven auto-assembly process of the water-oxidizing Mn_4_CaO_5_-cluster in photosystem II" for further consideration at *eLife*. Your revised article has been favorably evaluated by Philip Cole (Senior editor), a Reviewing editor, and two reviewers.

The manuscript has been improved but there are some remaining issues that need to be addressed before acceptance, as outlined below. In particular, the concerns of Reviewer 2 should be addressed prior to acceptance.

Reviewer #1:

The revised manuscript properly addresses the concerns raised by the referees. I think that it merits publication in *eLife* and it is going to positively impact the research of photosystem II that is one of the most important membrane complexes in Nature. I do hope that in the future the authors will aim for a complete reconstitution of the Mn cluster.

Reviewer #2:

This revised manuscript is improved, but additional questions remain regarding the reconstitution protocol and data.

Subsection “Partial reconstitution of the Mn_4_CaO_5_-cluster “("15-20 minutes of dark incubation") contradicts ("…15 minutes incubation time… the crystals were illuminated with green light…")

How long was the incubation? 15 minutes or 15-20 minutes? Was the incubation performed in darkness or with low intensity green light? Was this green light exposure continuous (e.g. incubation under the microscope) or intermittent (e.g. occasionally exposed to headlamp light during incubation)? What was the concentration of chlorophyll or PSII in the crystal suspension?

Please use the PSII concentration value, the extinction coefficient of PSII at the specific wavelength of green light used, and the exact duration of 2.6 μE m-2 s^-1^ green light exposure to estimate the number of photons absorbed by each reaction center. This "back of the envelope" calculation is needed to address the concern that the sample was effectively dark during incubation with manganese.

Technically, these details are important so that the experiments can be reproduced by other labs. For the current manuscript, these details dictate whether the authors observed dark reconstitution of a single manganese ion at the high affinity site or an actual WOC PHOTO-assembly intermediate containing two manganese ions.

---

## [Author Response]

Reviewer #1:

*Since George Cheniae's early discoveries in the early 1970's, it is known that hydroxylamine causes reduction of Mn in the manganese cluster of PSII followed by loss of oxygen evolution. In cyanobacteria, the loss of the functional manganese cluster results in the loss of the three extrinsic subunits PsbU, PsbV and PsbO following by practical disassembly of the dimeric PSII complex. Reactivation of PSII involves denovo synthesis and assembly of subunits, incorporation of lipids and pigments and finally assembly of the manganese cluster. The stepwise process of photo-activation involves the incorporation of the Mn and Ca ions into the dimeric PSII core complex. First, a Mn(II) ion binds in the dark and is oxidized after absorption of a first light quantum forming an unstable Mn(III) intermediate. Following a light independent rearrangement, a second light quantum drives the oxidation of a second Mn(II) ion resulting in a next assembly intermediate. The incorporation of the other two missing Mn and Ca ions to complete the metal-cluster is not clearly resolved yet.*

*Recently "Quebane"-like manganese clusters were synthesized demonstrating the stability and possible self-assembly of such complex. (Barber 2017). However the precise arrangement and the interdependency with the coordinating ligands is not settled. The novelty of this work is that PSII crystals were used for stabilizing the environment of the manganese cluster such that it could be totally depleted without alteration not only in the protein clusters but also of the bound water molecules in its vicinity. This is a remarkable achievement. However the characteristic EPR TyrD● signal was significantly weaker than the signal of the untreated sample presumably because the TyrD● radical was reduced after addition of NH2OH. This alteration might be one of the obstacles that prevented a complete restoration of the oxygen evolution. Even though only partial reconstitution of the manganese cluster was achieved there are sufficient new observations that were achieve at high proficiency to merit publication of the manuscript. This manuscript also can set example for many other unrelated studies that report crystal structures without taking advantage of experiments that could be performed using the crystals.*

We thank the reviewer for the positive comments and wanted to add that we did not attempt a full reconstitution of a functional Mn cluster in the present work. We rather intended to investigate the structure of the apo site and get some insight into binding of the first Mn in the photoactivation process. We plan to extend these studies in the future to later assembly intermediates and hopefully – eventually to a fully reassembled Mn cluster.

*Reviewer #2:*

*The authors describe their reconstitution protocol in subsection “Reconstitution of the Mn_4_CaO_5_-cluster**”**. However, not enough detail is provided. Was light provided? What is the occupancy of the QB site? Were electron acceptors added? If no light was used, then only one Mn(II) would be expected to bind to the high affinity site. Only if light was provided is it possible for more than one Mn to be observed.*

We have revised the description of the reconstitution protocol to provide more details (Materials and methods). No electron acceptors were added. The reconstitution experiment did not take place in the dark since we had to work with green light (< 2.6 μE /m^[2]^ /s) during the treatment of the crystals under the microscope. The accumulated low light exposure was almost certainly sufficient to activate the photooxidation process from Mn^2+^ to Mn^3+^ at the high affinity site of the WOC. At the current resolution of 4.5 Å, as achieved for the partially reconstituted crystals, the electron density map of the mobile plastoquinone molecule (Q_B_) does not allow to decide about the localization of Q_B_. This is only possible at higher resolutions, better than 3 Å (Loll et al., (2005) resolution. Hence, a PSII crystal structure at a resolution of at least 2.9 Å (Guskov et al., (2009). would be required to determine the occupancy of the Q_B_ site. On the other hand it is safe to assume that in the present preparations a sufficient amount of quinones is present to allow transfer of 1 or 2 electrons, as would be necessary for the initial photooxidation of one or two Mn. We previously showed for crystals from similar protein preparations that we can transfer up to 6 electrons without addition of artificial acceptors (Krivanek et al., (2007); (Young et al., (2016), indicating the presence of a small quinone pool in these crystals.

*In addition, the MnCl_2_ and CaCl_2_ concentrations used (10mM and 5mM, respectively) are not compatible with efficient photo-assembly. In photo-assembly studies by Cheniae, Burnap, Dismukes, and Brudvig, Ca^2+^ concentration is always much higher than Mn^2+^ (as is the case* in vivo*). Maintaining a high [Ca^2+^]/[Mn^2+^] ration is critical for preventing improperly formed Mn clusters containing >5Mn per PSII (see Chen, Cheniae 1995). The authors should show that their conditions promote a high yield of photo-assembly with ~4Mn per PSII.*

An optimal ratio of Mn^2+^ to Ca^2+^ was needed for a high yield of photoassembly. In earlier studies, a significantly higher concentration of Ca^2+^ was always used in comparison to Mn^2+^. Baranov et al., have determined a fixed Mn^2+^ to Ca^2+^ ratio (1:500) which produces the fastest rate and highest yield of photoactivation (Baranov et al., (2004). In contrast, using only 10 to 20- fold higher concentration of Ca^2+^ results in considerably slower rate of photoassembly (Kolling et al., (2012). However, it must be considered that the Mn^2+^/Ca^2+^ ratio suggested from the literature is always for photoassembly experiments with PSII-enriched thylakoid membrane fragments in solutions (e.g. Cheniae, Burnap, Dismukes and Brudvig). In contrast, we used suspensions of crystals of PSII core complex. During our studies we tested various concentrations of MnCl_2_ to find the optimum conditions for reconstitution. Whereas very low concentrations did not yield rebinding of Mn, we were able to observe a partial reassembly of the Mn_4_CaO_5_-cluster in PSII crystals at around 10mM MnCl_2._Since we treated the apo-PSII crystals in the presence of (NH_4)2_SO_4_, TRIS/HCl and PEG 5000 MME (conditions required to prevent the crystals from dissolving), we observed a precipitation of CaSO_4_ when increasing the Ca^2 +^ concentration beyond 5 mM. Therefore, we did not attempt a further increase of the concentration of Ca^2+ -^ions and used the maximum Ca concentration possible under our buffer conditions.

The first Photosystem II crystal structure (Zouni et al., (2001) was determined by a phase extension of 4.2 to 3.8 Å resolution by using a cadmium data set. In this work, we have shown that even small concentrations as well as short incubation times of metal ions (salts) suffice to diffuse into the PSII crystals. The PSII crystals were incubated with only 1 mM CdSO_4_ for a few minutes and a binding at the PsbO subunit was observed. Nevertheless, a clear identification of the Ca next to the Mn in the WOC was only possible in the electron density of PSII at 3.0 Å resolution due to its weaker electron density compared to the Mn ions (see discussion in Kern et al., (2007).

At the limited resolution of 4 - 4.5 Å it is even more difficult to see Ca compared to Mn, so that we cannot exclude that Ca is present. In addition, the Ca site could be not fully occupied or not well-ordered and therefore even more difficult to see compared to the Mn ions.

In this work, first structural data on the partial re-assembly of the Mn_4_CaO_5_-cluster of PSII were presented at a resolution of 4.5 Å and we want to emphasize that we did not attempt yet to reconstitute a fully activate Mn_4_CaO_5_-cluster. Further and detailed experiments involving different buffer conditions, variations of the ratio Ca^2 +^ / Mn^2+^ and different light conditions on apo-PSII microcrystals by means of membrane inlet mass spectroscopy are planned in the near future to establish optimum conditions for reconstitution of a fully active Mn cluster. But these studies are beyond the scope of the current work. We modified the main text accordingly; see revised text in subsection”Complete and partial Mn_4_CaO_5_-cluster depletion on dPSIIcc crystals”.

*It is surprising and concerning that electron density from Ca^2+^ is not observed during reconstitution. In subsection “Possible intermediates of disassembly and photo-activation**“,** the authors state "It is disputed whether the calcium ion binds at an early or late photo-activation state" and cite a review by Bao and Burnap. This statement is a mischaracterization of the review and is at odds with the EPR results from Tyryshkin, Dismukes, and coworkers (2006) that Ca^2+^ is present near the first mononuclear Mn3+ intermediate of photo-assembly. The absence of Ca^2+^ makes the possibility of an improperly assembled cluster more likely (see previous paragraph).*

We agree with the reviewer that the wording for this section was not carefully thought through and modified this section to better describe the literature on the Ca requirement, see subsection “Possible intermediates of disassembly and photo-activation”. As stated above we cannot exclude the presence of Ca based on our low resolution electron densities and we stated this more clearly in the text and also added a sentence to the legend of Figure 6. We are aware that there is the possibility of an improperly assembled cluster under low Ca^2+^ conditions but we did not observe any additional Mn binding (beyond the 1 or 2 that we describe in the manuscript) under our experimental conditions. Furthermore, we observe a similar electron density in terms of extension and position for both the partially disassembled and the partially reconstituted crystals, indicating that we likely observe a similar intermediate in both cases. As stated above the scope of the present work was not to reassemble a fully functional Mn cluster but rather to investigate the structure of the apo binding site and characterize the position of the first Mn bound to the apo site. Future studies will be conducted to characterize the intermediates of assembly in more detail and to find conditions for optimum reconstitution of the full Mn cluster.

[Editors' note: further revisions were requested prior to acceptance, as described below.]

*Reviewer #2:*

*This revised manuscript is improved, but additional questions remain regarding the reconstitution protocol and data.*

*Subsection “Partial reconstitution of the Mn_4_CaO_5_-cluster “("15-20 minutes of dark incubation") contradicts ("…15 minutes incubation time… the crystals were illuminated with green light…")*

*How long was the incubation? 15 minutes or 15-20 minutes? Was the incubation performed in darkness or with low intensity green light? Was this green light exposure continuous (e.g. incubation under the microscope) or intermittent (e.g. occasionally exposed to headlamp light during incubation)? What was the concentration of chlorophyll or PSII in the crystal suspension?*

The incubation times mentioned in subsection “Partial reconstitution of the Mn_4_CaO_5_-cluster” described two different facts. *("15-20 minutes of dark incubation")* referred to the reconstitution in contrast to *("…15 minutes incubation time… the crystals were illuminated with green light…")* which described the incubation during the dehydration process.

The incubation was performed in darkness, but the sequential treatments of crystals (i.e., buffer exchange steps) were carried out with low intensity of intermittent green light of the microscope (2.6µE/m^[2]^/s) yielding in a total illumination time of 20 minutes over the entire procedure. We slightly modified the text to clarify these details.

As we only incubated a few single crystals at a time the chlorophyll concentration of the suspension is not relevant. Rather the chlorophyll concentration and the thickness for a single PSII crystal (59 mM, 50 µm) need to be used to calculate the absorption by the sample during the illumination procedure.

*Please use the PSII concentration value, the extinction coefficient of PSII at the specific wavelength of green light used, and the exact duration of 2.6 μE m^-2^ s^-1^ green light exposure to estimate the number of photons absorbed by each reaction center. This "back of the envelope" calculation is needed to address the concern that the sample was effectively dark during incubation with manganese.*

The crystal size is 100 µm in length, 50 µm width and 50 µm thickness. The light intensity is 2.6 µE/m^[2]^/s and the exposure time of the crystals is 20 minutes over the entire procedure. We calculated 15 absorbed photons/reaction center by using an extinction coefficient of 5.8 mM^-1^ cm^-1^ at 532 nm.

The quantum efficiency is about one percent for the full activation of the Mn_4_CaO_5_-cluster. Assuming four steps about thirty percent quantum efficiency is necessary in each step. This leads to about three absorbed photons/RC being required for the first step. We calculated in the green light exposed (20 minutes) PSII crystal about 15 absorbed photons per RC. That means that the oxidation of Mn^2+^ to Mn^3+^ should take place under these conditions.

*Technically, these details are important so that the experiments can be reproduced by other labs. For the current manuscript, these details dictate whether the authors observed dark reconstitution of a single manganese ion at the high affinity site or an actual WOC PHOTO-assembly intermediate containing two manganese ions.*